# Differences in the immune response elicited by two immunization schedules with an inactivated SARS-CoV-2 vaccine in a randomized phase 3 clinical trial

Nicolás MS Gálvez[1,2†], Gaspar A Pacheco[1,2†], Bárbara M Schultz[1,2†], Felipe Melo-González[1,2,3†], Jorge A Soto[1,2,3†], Luisa F Duarte[1,2†], Liliana A González[1,2†], Daniela Rivera-Pérez[1,2], Mariana Ríos[1,2], Roslye V Berrios[1,2], Yaneisi Vázquez[1,2], Daniela Moreno-Tapia[1,2], Omar P Vallejos[1,2], Catalina A Andrade[1,2], Guillermo Hoppe-Elsholz[1,2], Carolina Iturriaga[4], Marcela Urzua[4], María S Navarrete[5], Álvaro Rojas[6], Rodrigo Fasce[7], Jorge Fernández[7], Judith Mora[7], Eugenio Ramírez[7], Aracelly Gaete-Argel[8], Mónica L Acevedo[1,8], Fernando Valiente-Echeverría[8], Ricardo Soto-Rifo[8], Daniela Weiskopf[9], Alba Grifoni[9], Alessandro Sette[9,10], Gang Zeng[11], Weining Meng[12], CoronaVacCL03 Study Group, José V González-Aramundiz[13], Marina Johnson[14], David Goldblatt[14], Pablo A González[1,2], Katia Abarca[1,4], Susan M Bueno[1,2*], Alexis M Kalergis[1,2,15*]

[1]Millennium Institute on Immunology and Immunotherapy, Santiago, Chile; [2]Departamento de Genética Molecular y Microbiología, Facultad de Ciencias Biológicas, Pontificia Universidad Católica de Chile, Santiago, Chile; [3]Departamento de Ciencias Biológicas, Facultad de Ciencias de la Vida, Universidad Andrés Bello, Santiago, Chile; [4]Departamento de Enfermedades Infecciosas e Inmunología Pediátrica, División de Pediatría, Escuela de Medicina, Pontificia Universidad Católica de Chile, Santiago, Chile; [5]Centro de Investigación Clínica UC, Pontificia Universidad Católica de Chile, Santiago, Chile; [6]Departamento de Enfermedades Infecciosas del Adulto, División de Medicina, Escuela de Medicina, Pontificia Universidad Católica de Chile, Santiago, Chile; [7]Departamento de Laboratorio Biomédico, Instituto de Salud Pública de Chile, Santiago, Chile; [8]Laboratory of Molecular and Cellular Virology, Virology Program, Institute of Biomedical Sciences, Faculty of Medicine, Universidad de Chile, Santiago, Chile; [9]Center for Infectious Disease and Vaccine Research, La Jolla Institute for Immunology, La Jolla, United States; [10]Department of Medicine, Division of Infectious Diseases and Global Public Health, University of California, San Diego, United States; [11]Sinovac Biotech, Beijing, China; [12]Sinovac Life Sciences Co., Ltd., Beijing, China; [13]Departamento de Farmacia, Facultad de Química y de Farmacia, Pontificia Universidad Católica de Chile, Santiago, Chile; [14]Department of Infection, Inflammation and Immunity, Great Ormond Street Institute of Child Health, University College London, London, United Kingdom; [15]Departamento de Endocrinología, Facultad de Medicina, Escuela de Medicina, Pontificia Universidad Católica de Chile, Santiago, Chile

*For correspondence:
sbueno@bio.puc.cl (SMB);
akalergis@bio.puc.cl (AMK)

†These authors contributed equally to this work

## Abstract

**Background:** The development of vaccines to control the coronavirus disease 2019 (COVID-19) pandemic progression is a worldwide priority. CoronaVac is an inactivated severe acute respiratory syndrome coronavirus 2 (SARS-CoV-2) vaccine approved for emergency use with robust efficacy and immunogenicity data reported in trials in China, Brazil, Indonesia, Turkey, and Chile.

**Methods:** This study is a randomized, multicenter, and controlled phase 3 trial in healthy Chilean adults aged ≥18 years. Volunteers received two doses of CoronaVac separated by 2 (0–14 schedule) or 4 weeks (0–28 schedule); 2302 volunteers were enrolled, 440 were part of the immunogenicity arm, and blood samples were obtained at different times. Samples from a single center are reported. Humoral immune responses were evaluated by measuring the neutralizing capacities of circulating antibodies. Cellular immune responses were assessed by ELISPOT and flow cytometry. Correlation matrixes were performed to evaluate correlations in the data measured.

**Results:** Both schedules exhibited robust neutralizing capacities with the response induced by the 0–28 schedule being better. No differences were found in the concentration of antibodies against the virus and different variants of concern (VOCs) between schedules. Stimulation of peripheral blood mononuclear cells (PBMCs) with Mega pools of Peptides (MPs) induced the secretion of interferon (IFN)-γ and the expression of activation induced markers in CD4+ T cells for both schedules. Correlation matrixes showed strong correlations between neutralizing antibodies and IFN-γ secretion.

**Conclusions:** Immunization with CoronaVac in Chilean adults promotes robust cellular and humoral immune responses. The 0–28 schedule induced a stronger humoral immune response than the 0–14 schedule.

**Funding:** Ministry of Health, Government of Chile, Confederation of Production and Commerce & Millennium Institute on Immunology and Immunotherapy, Chile.

**Clinical trial number:** NCT04651790

## Editor's evaluation

This manuscript investigates the humoral neutralizing antibody and cellular immune responses of volunteers in a randomized clinical trial for the CoronaVac SARS-CoV-2 vaccine. The findings are useful and provide context for the efficaciousness of the 0-14 day and 0-28 day dosing schedules of CoronaVac. The results show that these two dosing schedules are similar across most metrics.

## Introduction

The current coronavirus disease 2019 (COVID-19) pandemic is caused by severe acute respiratory syndrome coronavirus 2 (SARS-CoV-2) (*Kim et al., 2020*; *Zhu et al., 2020*), a virus described for the first time at Wuhan, China, in late December 2019. SARS-CoV-2 is already responsible for more than 530 million cases of infection and over 6 million deaths during the past 2 years (*Dong et al., 2020*). Worldwide efforts to develop effective vaccines against this virus have led to 10 vaccine prototypes approved for emergency use by the WHO, and over 8 billion vaccines administered to humans to date (*World Health Organization, 2022*). Most approved SARS-CoV-2 vaccines rely on a single viral component, namely the spike (S) protein or its receptor-binding domain (RBD), which could negatively impact the neutralizing capacities of antibodies induced if circulating VOCs, such as the Delta and Omicron, mutate those sequences (*World Health Organization, 2022*). Whole virus-inactivated platforms have been widely used throughout history to prevent diseases against other viruses (*Kyriakidis et al., 2021*). In the case of SARS-CoV-2, these vaccines contain a broader diversity of antigens than just the S protein. Therefore, they might be more suited to protect against emerging circulating variants (*Kyriakidis et al., 2021*; *Melo-González et al., 2021*).

CoronaVac is an inactivated SARS-CoV-2 vaccine approved by the WHO for emergency administration to humans and developed by Sinovac Life Sciences Co., Ltd. (*Gao et al., 2020*; *Mallapaty, 2021*). Phase 1/2 trials in China confirmed that this vaccine induces a robust immune response against SARS-CoV-2 (*Wu et al., 2021*; *Zhang et al., 2021*). These data led to the evaluation of this vaccine in phase 3 clinical trials in other countries, including Brazil, Indonesia, Turkey, and Chile. An efficacy of 83.5%

was reported for CoronaVac in Turkey and an effectiveness of 87.5% in Chile to prevent hospitalization due to COVID-19 for healthy adults (*Jara et al., 2021*; *Tanriover et al., 2021*).

In this article, we compare the immune response elicited in healthy Chilean adults immunized with two doses of CoronaVac separated by either 2 (0–14 schedule) or 4 weeks (0–28 schedule). Our results suggest that, although the neutralizing capacities of antibodies elicited by a 0–28 immunization schedule with CoronaVac in Chilean adults are more robust than those induced by a 0–14 schedule, overall, both immune responses are equivalent.

## Materials and methods
### Study design, randomization, masking, and volunteers
This clinical trial (clinicaltrials.gov NCT04651790) was conducted in Chile at eight different sites, six located in Santiago city (Metropolitan Region) and two in the V Region of Valparaiso. The study protocol adhered to the current Tripartite Guidelines for Good Clinical Practices, the Declaration of Helsinki, and local regulations and was approved by the Institutional Scientific Ethical Committee of Health Sciences of the Pontificia Universidad Católica de Chile (#200708006). The execution was approved by the Chilean Public Health Institute (#24204/20).

Recruited volunteers were adults aged ≥18 years, and informed consent was obtained upon enrollment. Volunteers received two doses of CoronaVac at day 0 and 2 (0–14) or 4 (0–28) weeks after the first immunization. Volunteers did not receive any payment for their participation. Study nurses oversaw the immunization and did not participate in any other study procedure. Inclusion and exclusion criteria were applied as previously reported (*Bueno et al., 2022*). Briefly, inclusion criteria considered being aged 18 and over, being able to understand and sign the informed consent form, and compliances with all study procedures and visits. Exclusion criteria considered mainly history of confirmed symptomatic SARS-CoV-2 infection, pregnancy, allergy to vaccine components, and immunocompromised conditions. Well-controlled medical conditions were allowed.

Randomization was performed with a sealed enveloped system integrated with an electronic Case Report Forms (eCRF) in the OpenClinica platform; 2302 volunteers were enrolled in this phase 3 clinical trial by April 9, 2021. Four-hundred and forty volunteers were part of the immunogenicity arm; 199 of these were part of the 0–14 schedule; and 241 were part of the 0–28 schedule (*Figure 1—figure supplement 1*). The mean age of the recruited volunteers was 40.4±11.8 for the 0–14 schedule and 39.2±10.3 for the 0–28 schedule. A total of 52.9% volunteers were female and 47.1% were male.

CoronaVac consists of 3 µg or 600SU of β-propiolactone-inactivated SARS-CoV-2 (strain CZ02) with aluminum hydroxide as an adjuvant in 0.5 mL (*Gao et al., 2020*). Sodium chloride, monosodium hydrogen phosphate, and disodium hydrogen phosphate are excipients, and water for injection is included as solvent. A study nurse administered ready-to-use syringes with 0.5 mL of CoronaVac intramuscularly in the deltoid area. Sera and PBMCs were isolated from blood obtained before administration of the first and the second dose and 2 and 4 weeks after the second dose for both immunization schedules.

### Procedures
For the isolation of sera, 20 mL of blood were collected in anticoagulant tubes and distributed in two tubes of 10 mL per volunteer (BD Vacutainer Clot Activator tubes #367896). Blood was allowed to clot for at least 1 hr at room temperature (RT). Samples were then centrifuged in a refrigerated centrifuge with a horizontal rotor at 1300× *g* for 10 min at 22°C. Serum was collected and stored at –80°C until use. Hemolyzed samples were rejected. For the isolation of PBMCs, blood was collected in three heparinized tubes (BD Vacutainer #367874, 10 mL) and stored at RT until processing. Samples were diluted with PBS (1:1) and centrifuged for 10 min at 1200× *g* (RT) in SepMate tubes (StemCell Technologies) with density-gradient medium (Lymphoprep). The plasma was then discarded and PBMCs were isolated by pouring them into a clean tube. Isolated PBMCs were washed twice with sterile PBS, counted, and cryopreserved in FBS (Industrial Biologicals) and 10% DMSO (Chem Cruz). All PBMC samples were stored in liquid nitrogen until use.

To assess the presence of anti-SARS-CoV-2 antibodies, blood samples obtained before the first dose (preimmune), before the second dose, 2 and 4 weeks after the second dose were analyzed. The quantitative measurement of human IgG antibodies against the RBD domain of the S1 protein

(S1-RBD) was determined using the RayBio COVID-19 (SARS-CoV-2) Human Antibody Detection Kit (Indirect ELISA method) (Cat #IEQ-CoVS1RBD-IgG) and through meso-scale discovery (MSD) immunoassays were performed as described previously (*Johnson et al., 2020*). Briefly, these kits consist of 96-well plates coated with the S1-RBD protein segment SARS-CoV-2. Sera samples were serially diluted starting at a 200-fold dilution until a 6400-fold dilution. After 1 hr of incubation at RT, the plates were washed, and a biotinylated anti-human IgG antibody provided in the kit was added and incubated for 30 min at RT. Plates were washed, and then a horseradish peroxidase (HRP)-conjugated streptavidin was added and incubated for 30 min at RT. Plates were rewashed, and the TMB substrate solution supplied in the kit was added. Finally, a stop solution provided in the kit was added, and absorbance was measured at 450 nm in an ELISA plate reader (Biotek, 1506021). As controls, dilutions of the First WHO International Standard for anti-SARS-CoV-2 immunoglobulin (human – NIBSC code: 20/13), and positive control provided by each kit were included. Additional controls included samples of volunteers seropositive or seronegative at recruitment or inoculated with placebo. To calculate the end titers, the cut-off value for each test was determined as $\geq$2.1 times the average $OD_{450nm}$ value of a panel of 8 serum samples from volunteers who received placebo and 12 seronegative serum samples. These serum samples were obtained before vaccination from seronegative volunteers. Seropositivity was determined as the highest dilution that reached >2.1 times the $OD_{450nm}$ cut-off value. Seroconversion was defined as an increase of at least four times the titer at baseline.

For the surrogate virus neutralization test (sVNT), the SARS-CoV-2 Neutralizing Antibodies Test (BSNAT) Kit from BioHermes (COV-S41) was used to detect neutralizing antibodies in serum against SARS-CoV-2. This kit is a blocking assay which simulates the neutralization process, based on the ELISA platform. The main components of the kit are an ELISA plate pre-coated with the human ACE2 (hACE2) protein and the SARS-CoV-2 S1-RBD fragment conjugated with horseradish peroxidase (HRP-RBD). Briefly, the first step consisted in preparing serial dilutions from serum samples with the sample dilutor provided in the kit. Then, samples and controls were incubated with the HRP-RBD for 10 min at 37°C to allow the binding of neutralizing antibodies to RBD. Sera and controls previously incubated with HRP-RBD were added to ELISA plates (pre-coated with hACE2) and incubated for 20 min at 37°C. After incubation, samples were discarded, and all the wells were washed five times. Finally, TMB solution was added and quenched after 15 min of incubation at RT. Plates were read at 450 nm in a microplate reader (EPOCH, Biotek 1506021). The neutralizing antibody titer was determined as the last fold dilution with a cut-off value over 30% of inhibition. The inhibition rate was calculated based on the negative control absorbances (negative control – sample/negative control*100), and 10% was considered as the cut-off value. A similar methodology was performed to evaluate the neutralizing antibodies in sera against different VOCs, as reported previously (*Melo-González et al., 2021*).

For the conventional virus neutralization test (cVNT), Vero E6 cells were infected with a SARS-CoV-2 strain obtained by viral isolation in tissue cultures (33782CL-SARS-CoV-2 strain). Neutralization assays were carried out by the reduction of cytopathic effect (CPE) in Vero E6 cells (ATCC CRL-1586, confirmed to be *Mycoplasma* free). The titer of neutralizing antibodies was defined as the highest serum dilution that neutralized virus infection, at which the CPE was absent compared with the virus control wells (cells with CPE). Vero E6 cells ($4\times10^4$ cells/well) were seeded in 96-well plates. For neutralization assays, 100 µL of 33782CL-SARS-CoV-2 (at a dose of 100 $TCID_{50}$) were incubated with serial dilutions of heat-inactivated sera samples (dilutions of 1:4, 1:8, 1:16, 1:32, 1:64, 1:128, 1:256, and 1:512) from volunteers for 1 hr at 37°C. Then, the mix was added to the 96-well plates with the Vero E6 cells. CPE on Vero E6 cells was analyzed 7 days after infection. For each test, a serum sample from uninfected patients (negative control) and a neutralizing COVID-19 patient serum sample (positive control) were used.

For the pseudotyped virus neutralization test (pVNT), anti-SARS-CoV-2 neutralizing antibodies were measured using an HIV-1 backbone expressing firefly luciferase as a reporter gene and pseudotyped with the SARS-CoV-2 spike glycoprotein (HIV-1-SΔ19) as previously described (*Beltrán-Pavez et al., 2021*). Briefly, serum samples were initially diluted 1:4 in DMEM, serially diluted 1:3 up to 1:8748, and then mixed with approximately 4.5 ng of p24 equivalents of HIV-1-SΔ19 in white 96-well plate. Plates were incubated for 1 hr at 37°C, and then 100 mL of DMEM containing $1\times10^4$ HEK-ACE2 (transformed from CRL-3216 ATCC, confirmed to be *Mycoplasma* free) cells was added to each well. Firefly luciferase activity was measured 48 hr later, using the Luciferase Assay System (Promega) in a Glomax-96 microplate luminometer (Promega). Estimation of the ID50 was obtained using a

four-parameter nonlinear regression curve fit measured as the percent of neutralization determined by the difference in average relative light units between test samples and pseudotyped virus controls as previously described (*Beltrán-Pavez et al., 2021*). Data analyses and statistical analyses were carried out using GraphPad Prism v.9.

To assess the cellular immune response, ELISPOT and flow cytometry assays were performed using PBMCs from volunteers at the different times indicated above. Upon thawing, cells were resuspended in fresh media in a 1:10 dilution to remove DMSO remnants from the freezing media. Then, cells were centrifuged, resuspended in fresh media, and counted in an automated cell counter (Logos Biosystems #L40001). Cells were adjusted to $6 \times 10^6$ cells/mL and kept at 37°C, 5% $CO_2$ for 15 min until use in the corresponding assay. ELISPOT plates containing a PVDF membrane were activated with 15 µL of 70% ethanol (Merck), washed three times with sterile ×1 PBS, and then coated with human IFN-γ and IL-4 capture antibodies (1:250 and 1:125, respectively, CTL). After 3 hr of activation at RT, plates were washed two times with PBS and two times with PBS-Tween 20 0.05%. The stimulus included in these assays considers the use of Mega Pools (MPs) of peptides derived from SARS-CoV-2 proteins, previously described (*Grifoni et al., 2020*). Two MPs composed of peptides from the S protein (MP-S) and the remaining proteins of the viral particle (MP-R) were used, as previously described (*Grifoni et al., 2020*). These peptides were determined in silico to stimulate CD4+ T cells optimally. Also, two MPs composed of peptides from the proteome of SARS-CoV-2 (CD8-A and CD8-B) were used, as previously described (*Grifoni et al., 2020*). These peptides were determined in silico to optimally stimulate CD8+ T cell. As positive controls, an independent stimulation performed with 5 mg/mL of Concanavalin A (ConA) (Sigma Life Science #C5275-5MG), and with an MP of peptides derived from cytomegalovirus proteins (MP-CMV) for the stimulation of both CD4+ and CD8+ T cells (*Grifoni et al., 2020*). As a vehicle control, DMSO 1% (Merck #317275) was included. A total of $3 \times 10^5$ cells in 50 µL of media were added to each well containing 50 µL of media with the corresponding stimulus. The final concentration of each stimulus per well was 1 µg/mL (except for ConA and DMSO). Positive controls for ELISPOT assays considered $5 \times 10^4$ cells/well instead of $3 \times 10^5$ cells/well. For ELISPOT assays, cells were incubated for 48 hr at 37°C, 5% $CO_2$. After incubation, plates were washed one time with PBS, and three times with PBS-Tween 20. Then, anti-human IFN-γ (FITC) and anti-human IL-4 (Biotin) antibodies (1:1000 and 1:1000, respectively) were added, and plates were incubated for 2 hr, RT. Plates were washed three more times with PBS-Tween 20 and then FITC-HRP and Streptavidin-AP (1:1000) were added and plates were incubated for 1 hr, RT. After incubation, plates were washed three more times with PBS-Tween 20. Then, plates were treated with the blue (15 min), and red (15 min) developer solution individually following the recommendations of the manufacturer. Plates were washed with tap water after each developer solution and allowed to dry for 24 hr prior to reading. To evaluate the number of T cells secreting IFN-γ, IL-4, or both, ELISPOT assays were performed with ImmunoSpot technology (ImmnunoSpot #hIFNgIL4-1M-10). Spot forming cells (SFCs) were counted on an ImmunoSpot S6 Micro Analyzer.

To characterize the expression of activation-induced markers (AIM) by T cells, flow cytometry assays were performed. $3 \times 10^5$ cells/well were stimulated as described for the ELISPOT assays, and after 24 hr of incubation with the stimulus, samples were stained. Staining was performed by incubation for 45 min at 4°C using the following reagents: a mix of BD Horizon Fixable Viability Stain 510 (BD Biosciences – CAT 564406–1 µL per $1 \times 10^6$ cells); anti-CD14 V500 (BD Biosciences – clone M5E2); anti-CD19 V500 (BD Biosciences – clone HIB19); anti-CD3 AF-700 (Biolegend – clone OKT3); anti-CD69 PE (BD Biosciences – clone FN50); anti-CD8a BV-650 (BD Biosciences – clone RPA-T8); anti-CD4 BV-605 (BD Biosciences – clone RPA-T4); anti-CD137 (BioLegend, Clone 4-1BB); and anti-OX40 (BioLegend, Clone BER-ACT35). Cells were washed twice with 200 µL of PEB buffer, fixed, and then handed to the Flow Cytometry core facility, for their acquisition in an LSRFortessa X-20 flow cytometer.

## Statistical analyses

Sample size determination was already reported for this trial (*Bueno et al., 2022*). Statistical significance was set at α=0.05 in all cases. Statistical analyses and symbols used for each analysis are described briefly in each figure legend. All statistical analyses were performed in GraphPad Prism v.9.0.1 or RStudio.

To evaluate statistical differences of anti-S antibody titers and neutralizing antibody titers either by cVNT, sVNT, or pVNT induced by either immunization schedule, a two-tailed one-way ANOVA

for repeated measures was performed over the $Log_2$ of antibody titers, followed by Bonferroni's multiple comparisons test in order to compare both schedules. Differences in seroconversion rates of neutralizing antibody titers were determined by a two-tailed Fisher's exact test. Volunteers that were detected to be seropositive at entry for anti-N and/or anti-S antibodies were excluded from the analyses.

For the ELISPOT data, comparison between schedules was performed by a two-tailed one-way ANOVA for repeated measures which was performed over the $Log_2$ of antibody titers, followed by Bonferroni's multiple comparisons test over the $Log_{10}$ of the fold change. Differences among schedules for flow cytometry data were assessed by a two-tailed one-way ANOVA for repeated measures which was performed over the $Log_2$ of antibody titers, followed by Bonferroni's multiple comparisons test of the percentage data.

Pearson correlation matrixes were generated for each immunization schedules considering both humoral and cellular immune response data. Humoral response data were transformed to the base 10 logarithm before analysis. Individual Pearson correlations were selected based on their immunological relevance, n, r, and p values. These values, as well as the 95% confidence bands of the correlation, are shown in each graph.

Other comparisons assessed were analyzed via two-tailed unpaired t tests, two-tailed nonparametric unpaired t tests (Mann-Whitney tests), two-tailed non-parametric paired t tests (Wilcoxon tests), ANOVA followed by Tukey tests, or two-tailed Fisher's exact test, as indicated in the figure legends.

## Results

### A 0–28 day immunization schedule with CoronaVac promotes higher seropositivity rates and GMT values of neutralizing antibodies than a 0–14 schedule

To evaluate the humoral immune response elicited after vaccination with two doses of CoronaVac, separated by 2 or 4 weeks (0–14 and 0–28 days schedules, respectively), the neutralizing capacities of circulating antibodies were evaluated. This was performed independently through an sVNT (*Figure 1A and C*) and cVNT for the Ancestral strain (*Figure 1B and D*), as well as a pseudotyped virus neutralization test (pVNT) (*Figure 1—figure supplement 2*). For both immunization schedules, samples from 130 volunteers were tested for sVNT, 372 volunteers for cVNT with the Ancestral strain, and 94 for pVNT (*Table 1*). These techniques show a robust increase in arbitrary WHO international units (IU), geometric mean titer (GMT) values, and seropositivity rates 2 and 4 weeks after the second dose for both immunization schedules. Remarkably, as seen in IU for the sVNT and GMT values for cVNT and pVNT, the 0–28 schedule showed increased neutralizing capacities 2 and 4 weeks after the second dose (*Figure 1A, B* and *Figure 1—figure supplement 2A*). No differences in seropositivity rates between both schedules were detected for any of the assays evaluated (*Figure 1C, D* and *Figure 1—figure supplement 2B*). We also evaluated differences in the neutralizing capacities of circulating antibodies between the two age groups indicated for all four techniques (*Figure 1—figure supplement 3*). Both age groups had significantly increased GMT values at all times compared to preimmune samples, irrespective of the immunization schedule and the technique evaluated (*Table 2*). However, we observed significantly higher titers of circulating neutralizing antibodies in the 18–59 years age group compared to the >60 years age group, 2 and 4 weeks after the second dose, irrespective of the vaccination schedule, as determined by cVNT and pVNT (*Figure 1—figure supplement 3* and *Table 1*). These results suggest that CoronaVac induces the production of circulating antibodies with varying neutralizing capacities after immunization with either a 0–14 or a 0–28 schedule. Remarkably, the 0–28 schedule promotes higher seroconversion rates and GMT or IU values of these neutralizing antibodies than the 0–14 schedule, as determined by sVNT and pVNT.

### Both immunization schedule with CoronaVac exhibits similar levels of anti-S1 and anti-RBD-specific antibodies

To evaluate the humoral immune response elicited after vaccination with two doses of CoronaVac, separated by 2 or 4 weeks (0–14 and 0–28 day schedules, respectively), antibody titers against the Ancestral S1 and the RBD of SARS-CoV-2 were evaluated before administration of the first and second

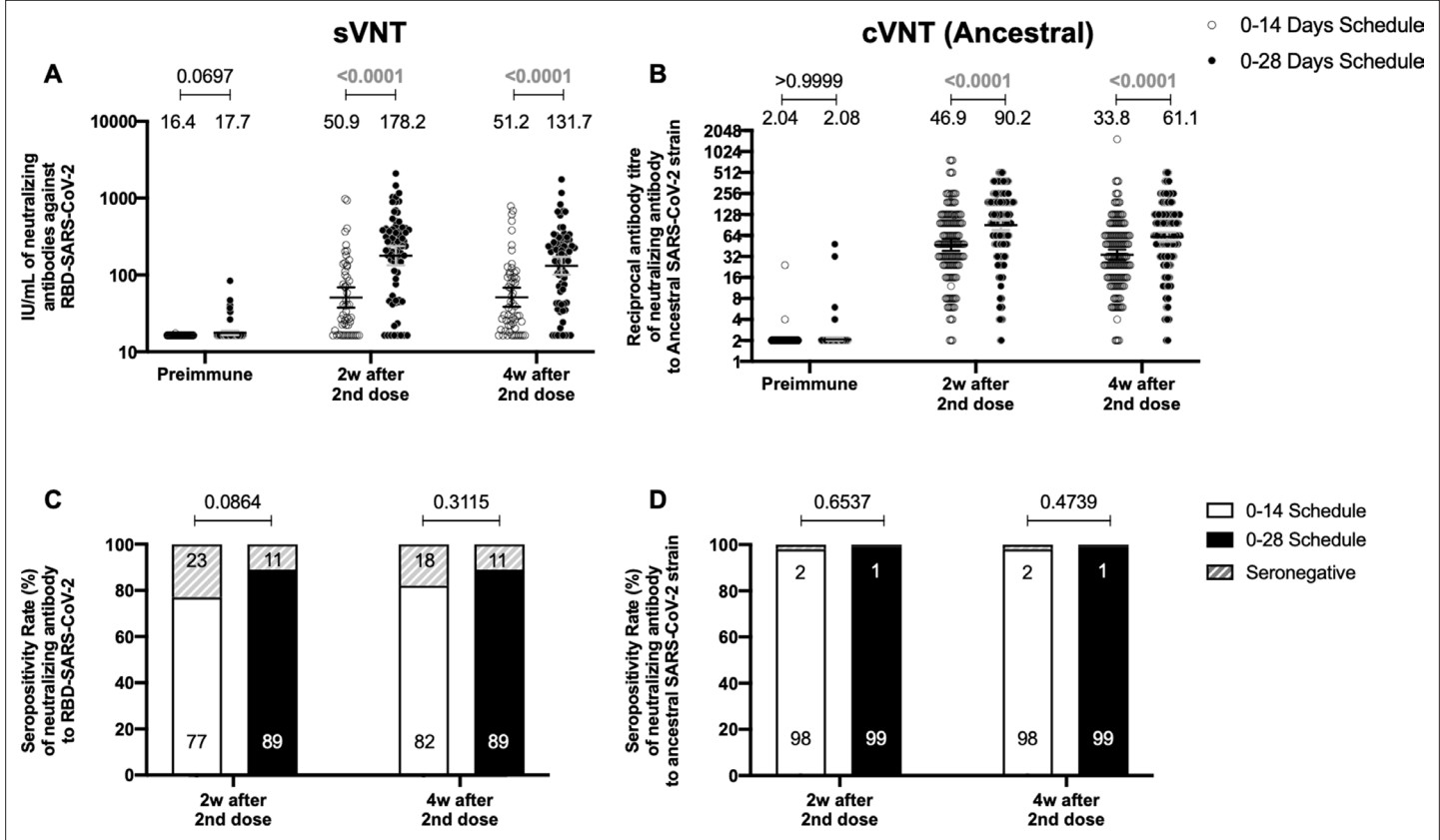

**Figure 1.** Circulating neutralizing antibodies against severe acute respiratory syndrome coronavirus 2 (SARS-CoV-2) measured by surrogate virus neutralization test (sVNT) and conventional virus neutralization test (cVNT) for the Ancestral strain in immunized volunteers. Neutralizing antibody titers were evaluated with an sVNT, which quantifies the interaction between S1-RBD and human ACE2 (hACE2) pre-coated on ELISA plates (**A,C**) and with a cVNT, which quantifies the cytopathic effect (CPE) induced in Vero cells as plaques formation (**B, D**). n=372 volunteers for cVNT (Ancestral) and n=130 volunteers for sVNT (for both schedules). Data is represented as the reciprocal antibody titer of neutralizing antibody versus the different times evaluated. Numbers above the bars show either the arbitrary international units (IU) (**A**) or the geometric mean titer (GMT) (**B**), and the error bars indicate the 95% CI. Seropositivity rates are also displayed (**C, D**). Data from IU and GMT values were analyzed by a two-tailed unpaired t-test of the base 2 logarithms of data to compare immunization schedules. Data from seropositivity rates were analyzed by a two-tailed Fisher's exact test. Numbers above each bracket represent calculated p values comparing both immunization schedules. Statistical significance was set at p<0.05 and highlighted numbers indicate statistical significance.

The online version of this article includes the following source data and figure supplement(s) for figure 1:

**Source data 1.** Data used to generate *Figure 1*, *Figure 1—figure supplement 3*, and *Figure 5*.

**Figure supplement 1.** Study design for this phase 3 clinical trial comparing two different immunization schedules as of August 2021.

**Figure supplement 2.** Geometric mean titer (GMT) values and seropositivity rates of circulating neutralizing antibodies against severe acute respiratory syndrome coronavirus 2 (SARS-CoV-2) measured through pseudotyped virus neutralization test (pVNT) (ID80).

**Figure supplement 2—source data 1.** Data used to generate *Figure 1—figure supplement 2*.

**Figure supplement 3.** Circulating neutralizing antibodies against severe acute respiratory syndrome coronavirus 2 (SARS-CoV-2) measured through surrogate virus neutralization test (sVNT), conventional virus neutralization test (cVNT), and pseudotyped virus neutralization test (pVNT) (ID80) in volunteers immunized with CoronaVac aged 18–59 and ≥60 years.

dose, and 2 and 4 weeks after the second dose (*Figure 2*). Samples from 162 volunteers were assessed independently for the S1-RBD through ELISA assays (*Figure 2A*) and 44 through MSD immunoassays (*Figure 2B*). Circulating antibodies against the S1-RBD were robustly increased for both immunization schedules at all times evaluated after administration of the first dose (preimmune), as determined by geometric mean units (GMUs) values of the arbitrary WHO international standard. No differences were found at all times evaluated for anti-S1-RBD-specific antibodies between both schedules (*Figure 2A*). Accordingly, no differences could be found between schedules for the MSD analyses performed, either for the S protein (left) or the RBD (right) of the Ancestral strain of SARS-CoV-2 (*Figure 2B*).

**Table 1.** Seropositivity rates and geometric mean titer (GMT) values measured for circulating neutralizing antibodies induced by CoronaVac in both immunization schedules and dissected by age group.

| Antibodies evaluated | Schedule | Age group | Indicators | 2 Weeks after second dose | *p Value | 4 Weeks after second dose | †p Value |
|---|---|---|---|---|---|---|---|
| sVNT | | | Seropositivity n/N | 41/53 | | 47/57 | |
| | | | % | 77.4 | | 82.5 | |
| | | | IU/mL | 50.9 | | 51.2 | |
| | | Total vaccine | 95% CI | 37.4–59.1 | | 38.5–68.0 | |
| | | | Seropositivity n/N | 30/32 | | 29/32 | |
| | | | % | 93.8 | | 90.6 | |
| | | | IU/mL | 76.0 | | 68.8 | |
| | | 18–59 years | 95% CI | 50.2–115.1 | | 46.0–102.9 | |
| | | | Seropositivity n/N | 11/21 | | 18/25 | |
| | | | % | 52.3 | | 72.0 | |
| | | | IU/mL | 27.6 | | 35.1 | |
| | 0–14 days | ≥60 years | 95% CI | 20.0–38.1 | 0.018 | 24.2–50.8 | 0.1100 |
| | 0–28 days | | Seropositivity n/N | 66/73 | 0.035 | 66/73 | 0.1100 |
| | | | % | 90.4 | | 90.4 | |
| | | | IU/mL | 178.2 | | 131.7 | |
| | | Total vaccine | 95% CI | 133.5–238.1 | | 100.7–172.3 | |
| | | | Seropositivity n/N | 33/33 | | 33/33 | |
| | | | % | 100 | | 100 | |
| | | | IU/mL | 255.9 | | 178.2 | |
| | | 18–59 years | 95% CI | 175.3–373.7 | | 123.6–256.9 | |
| | | ≥60 years | Seropositivity n/N | 33/40 | | 33/40 | |
| | | | % | 82.5 | | 82.5 | |
| | | | IU/mL | 132.0 | | 102.6 | |
| | | | 95% CI | 87.0–200.5 | | 70.0–150.3 | |

*Table 1 continued on next page*

Table 1 continued

| Antibodies evaluated | Schedule | Age group | Indicators | 2 Weeks after second dose | *p Value | 4 Weeks after second dose | †p Value |
|---|---|---|---|---|---|---|---|
| cVNT (Ancestral) | | | Seropositivity n/N | 147/150 | | 156/160 | |
| | | | % | 98.0 | | 97.5 | |
| | | | GMT | 46.9 | | 33.8 | |
| | | Total vaccine | 95% CI | 38.6–57.0 | | 28.4–40.2 | |
| | | | Seropositivity n/N | 121/121 | | 128/128 | |
| | | | % | 100 | | 100 | |
| | | | GMT | 59.2 | | 42.6 | |
| | | 18–59 years | 95% CI | 48.8–71.8 | | 35.7–50.9 | |
| | | | Seropositivity n/N | 26/29 | | 28/32 | |
| | | | % | 89.7 | | 87.5 | |
| | | | GMT | 17.3 | | 13.1 | |
| | 0–14 days | ≥60 years | 95% CI | 10.7–28.0 | <0.0001 | 8.9–19.1 | <0.0001 |
| | 0–28 days | | Seropositivity n/N | 208/210 | <0.0001 | 209/212 | <0.0001 |
| | | | % | 99.0 | | 98.6 | |
| | | | GMT | 90.2 | | 61.1 | |
| | | Total vaccine | 95% CI | 76.8–105.8 | | 52.3–71.4 | |
| | | | Seropositivity n/N | 124/125 | | 123/124 | |
| | | | % | 99.2 | | 99.2 | |
| | | | GMT | 121.9 | | 82.4 | |
| | | 18–59 years | 95% CI | 102.5–144.9 | | 69.6–97.7 | |
| | | ≥60 years | Seropositivity n/N | 84/85 | | 86/88 | |
| | | | % | 98.8 | | 97.7 | |
| | | | GMT | 57.9 | | 40.1 | |
| | | | 95% CI | 43.7–76.6 | | 30.7–52.4 | |

Table 1 continued on next page

*Table 1 continued*

| Antibodies evaluated | Schedule | Age group | Indicators | 2 Weeks after second dose | *p Value | 4 Weeks after second dose | †p Value |
|---|---|---|---|---|---|---|---|
| | | | Seropositivity n/N | 73/77 | | 73/77 | |
| | | | % | 97.3 | | 97.3 | |
| | | | GMT | 52.7 | | 40.1 | |
| | | Total vaccine | 95% CI | 36.6–76.4 | | 28.9–55.9 | |
| | | | Seropositivity n/N | 48/49 | | 48/49 | |
| | | | % | 97.9 | | 97.9 | |
| | | | GMT | 83.3 | | 59.0 | |
| | | 18–59 years | 95% CI | 53.6–129.5 | | 39.3–88.7 | |
| | | | Seropositivity n/N | 25/28 | | 25/28 | |
| | | | % | 89.2 | | 89.2 | |
| | | | GMT | 23.1 | | 20.0 | |
| | 0–14 days | ≥60 years | 95% CI | 13.2–40.7 | <0.0001 | 12.3–32.5 | 0.0027 |
| | | | Seropositivity n/N | 16/17 | | 16/17 | |
| | | | % | 94.1 | | 94.1 | |
| | | | GMT | 146.7 | | 104.9 | |
| | | Total vaccine | 95% CI | 60.0–359.0 | | 41.9–262.6 | |
| | | | Seropositivity n/N | 8/8 | | 8/8 | |
| | | | % | 100 | | 100 | |
| | | | GMT | 505.9 | | 328.7 | |
| | | 18–59 years | 95% CI | 306.1–836.0 | | 159.6–676.8 | |
| | | | Seropositivity n/N | 8/9 | | 8/9 | |
| | | | % | 88.8 | | 88.8 | |
| | | | GMT | 48.8 | | 38.0 | |
| pVNT | 0–28 days | ≥60 years | 95% CI | 13.3–178.6 | 0.0008 | 9.56–152.0 | 0.0029 |

Red values indicate statistically significant results (p<0.05).

*p Values are for comparison of IU/mL or GMT levels between 18 and 59 years and >60 years age groups 2 weeks after the second dose.

†p Values are for comparison of IU/mL or GMT levels between 18 and 59 years and >60 years age groups 4 weeks after the second dose.

These results show that CoronaVac induces a statistically significant increase in anti-S1 and anti-RBD antibodies after immunization with either a 0–14 or a 0–28 schedule.

## CoronaVac induces a significant cellular immune response against SARS-CoV-2 antigens regardless of the immunization schedule

To assess the cellular-mediated immune response elicited in volunteers immunized with CoronaVac in both immunization schedules, we evaluated the number of SFCs positive for IFN-γ by ELISPOT and the expression of AIM on T cells by Flow Cytometry (*Figure 3*). Peripheral blood mononuclear cells (PBMCs) from 88 volunteers were evaluated for both immunization schedules and techniques. To evaluate SARS-CoV-2 antigen-specific secretion of IFN-γ and expression of AIM by T cells, PBMCs were stimulated independently with four MPs of peptides comprising the proteome of SARS-CoV-2. One MP contains peptides from the S protein (MP-S, 15-mer peptides), and another one considers the remaining viral proteome (MP-R, 15-mer peptides). The two other MPs comprise the whole proteome of SARS-CoV-2. These MPs were split in two as they were too many to be used as a single stimulus (MP-CD8A and MP-CD8B, 9- to 11-mer peptides). Stimulation of PBMCs with MP-S and MP-R

**Table 2.** p Values estimated for neutralization assays evaluated for both immunization schedules.

| Figure | Parameter evaluated | Schedule; preimmune value | Preimmune compared to | |
|---|---|---|---|---|
| | | | 2 Weeks after second dose | 4 Weeks after second dose |
| *Figure 1A* | sVNT (IU/mL; p value) | 0–14 days; 16.4 | 50.9; <0.0001 | 51.2; <0.0001 |
| | | 0–28 days; 17.7 | 178.2; <0.0001 | 131.7; <0.0001 |
| *Figure 1B* | cVNT (Ancestral) (GMT; p value) | 0–14 days; 2.04 | 46.9; <0.0001 | 33.8; <0.0001 |
| | | 0–28 days; 2.08 | 90.2; <0.0001 | 61.1; <0.0001 |
| *Figure 1—figure supplement 2* | pVNT (GMT; p value) | 0–14 days; 2.05 | 52.66; <0.0001 | 40.14; <0.0001 |
| | | 0–28 days; 2.0 | 146.71; <0.0001 | 104.93; <0.0001 |

*p Values were determined by performing one-way ANOVAs for repeated measures over $Log_{10}$ of data, followed by post hoc Bonferroni's multiple comparisons test. Red values indicate statistically significant results (p<0.05).

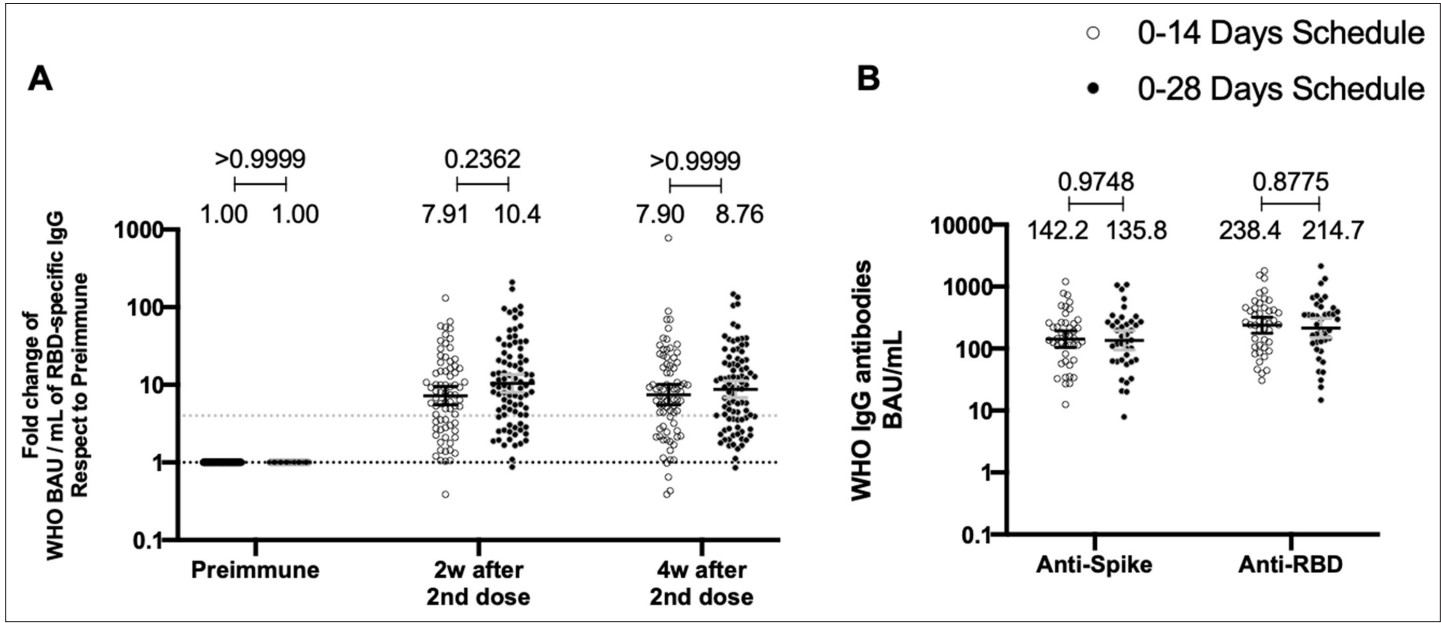

**Figure 2.** Total anti-S1 and anti-RBD antibodies circulating in immunized volunteers. Concentrations of IgG antibodies after two doses of CoronaVac were evaluated for immunized volunteers before the first (preimmune) and second dose and 2 and 4 weeks after the second. Specific IgG against the S1-RBD and the spike protein of severe acute respiratory syndrome coronavirus 2 (SARS-CoV-2) were measured. n=162 volunteers for ELISA assays (**A**) and n=44 volunteers for meso-scale discovery (MSD) assays (**B**). Data are expressed as the reciprocal antibody titer in arbitrary WHO international unit versus the different times evaluated. Error bars indicate the 95% CI. Spots represent individual values of each volunteer, with the numbers above each set of spots showing the geometric mean unit (GMU) estimates. Data were analyzed using a two-tailed unpaired t-test of the $Log_2$ of data to compare immunization schedules. Numbers above each bracket represent calculated p values comparing both immunization schedules. Statistical significance was set at p<0.05 . Dotted line on A is showing a value of 4, which is the threshold established for the seroconversion rate of each volunteer. Therefore, every spot over the dotted line represents volunteers that were considered positive for seroconversion relative to their preimmune sample.

The online version of this article includes the following source data for figure 2:

**Source data 1.** Data used to generate *Figure 2*.

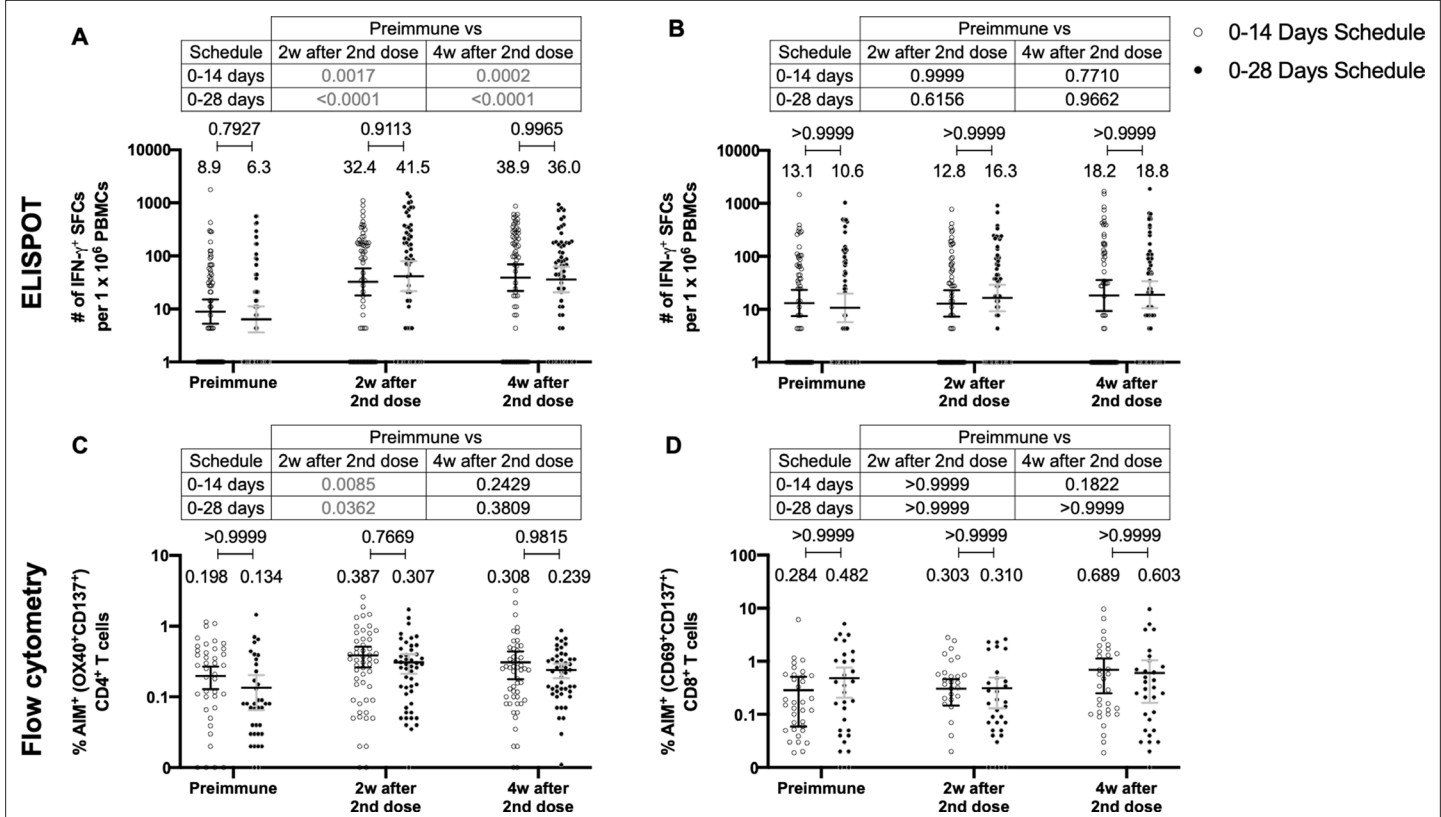

**Figure 3.** Similar levels of interferon (IFN)-γ-secreting cells and expression of activation-induced markers (AIM) on T cells are found upon stimulation with Mega Pools of peptides derived from severe acute respiratory syndrome coronavirus 2 (SARS-CoV-2) for both immunization schedules with CoronaVac. Total number of IFN-γ$^+$ spot forming cells (SFCs) were determined by ELISPOT. Data were obtained upon stimulation of peripheral blood mononuclear cells (PBMCs) for 48 hr with MP-S and -R (**A**) or with MP-CD8A and -B (**B**). The percentage of activated CD4$^+$ (AIM$^+$ [OX40$^+$, CD137$^+$]) and CD8$^+$ (AIM$^+$ [CD69$^+$, CD137$^+$]) T cells was determined by flow cytometry, upon stimulation for 24 hr with MP-S and -R (**C**), or with MP-CD8A and -B (**D**) in samples obtained before the first (preimmune) and second dose, and 2 and 4 weeks after the second dose. n=124 samples stimulated with MP-S and -R for ELISPOT (**A**). n=117 samples stimulated with MP-CD8A and -B for ELISPOT (**B**). n=116 stimulated with MP-S and -R for flow cytometry (**C**). n=110 samples stimulated with MP-CD8A and -B for flow cytometry (**D**) (for both schedules). Numbers above the bars show the mean and the error bars correspond to the 95% CI. Data were analyzed by a mixed-effect two-way ANOVA, followed by a Bonferroni's post hoc test to compare immunization schedules. Numbers above each bracket represent calculated p values comparing both immunization schedules. Statistical significance was set at p<0.05 and highlighted numbers indicate statistical significance.

The online version of this article includes the following source data and figure supplement(s) for figure 3:

**Source data 1.** Data used to generate *Figure 3*, *Figure 3—figure supplements 1–3*.

**Figure supplement 1.** Total number of interferon (IFN)-γ$^+$ spot forming cells (SFCs) induced upon stimulation with Mega Pools (MPs) of peptides derived from severe acute respiratory syndrome coronavirus 2 (SARS-CoV-2) proteome in volunteers immunized with CoronaVac aged 18–59 and ≥60 years.

**Figure supplement 2.** Percentage of activation-induced markers (AIM$^+$) T cells induced upon stimulation with Mega Pools (MPs) of peptides derived from severe acute respiratory syndrome coronavirus 2 (SARS-CoV-2) proteome in volunteers immunized with CoronaVac aged 18–59 and ≥60 years.

**Figure supplement 3.** Immunization with CoronaVac in a 0–28 schedule does not induce major IL-4 responses in peripheral blood mononuclear cells (PBMCs).

**Figure supplement 3—source data 1.** Data used to generate *Figure 3—figure supplement 3* and *Figure 5*.

induced a statistically significant increase in the secretion of IFN-γ and the expression of AIM in CD4$^+$ T cells (defined as OX40$^+$ and CD137$^+$), compared to preimmune samples (*Figure 3A and C*, tables on top of each panel). This increase was not detected when stimulating with MP-CD8A and MP-CD8B (*Figure 3B and D*, tables on top of each panel). No statistical differences could be found between both immunization schedules in the total numbers of IFN-γ$^+$ SFC (*Figure 3A, B*), or the percentage of AIM$^+$ T cells (*Figure 3C and D*). No differences between both immunization schedules were found when evaluating the age groups indicated before for SFC (*Figure 3—figure supplement 1*) or AIM

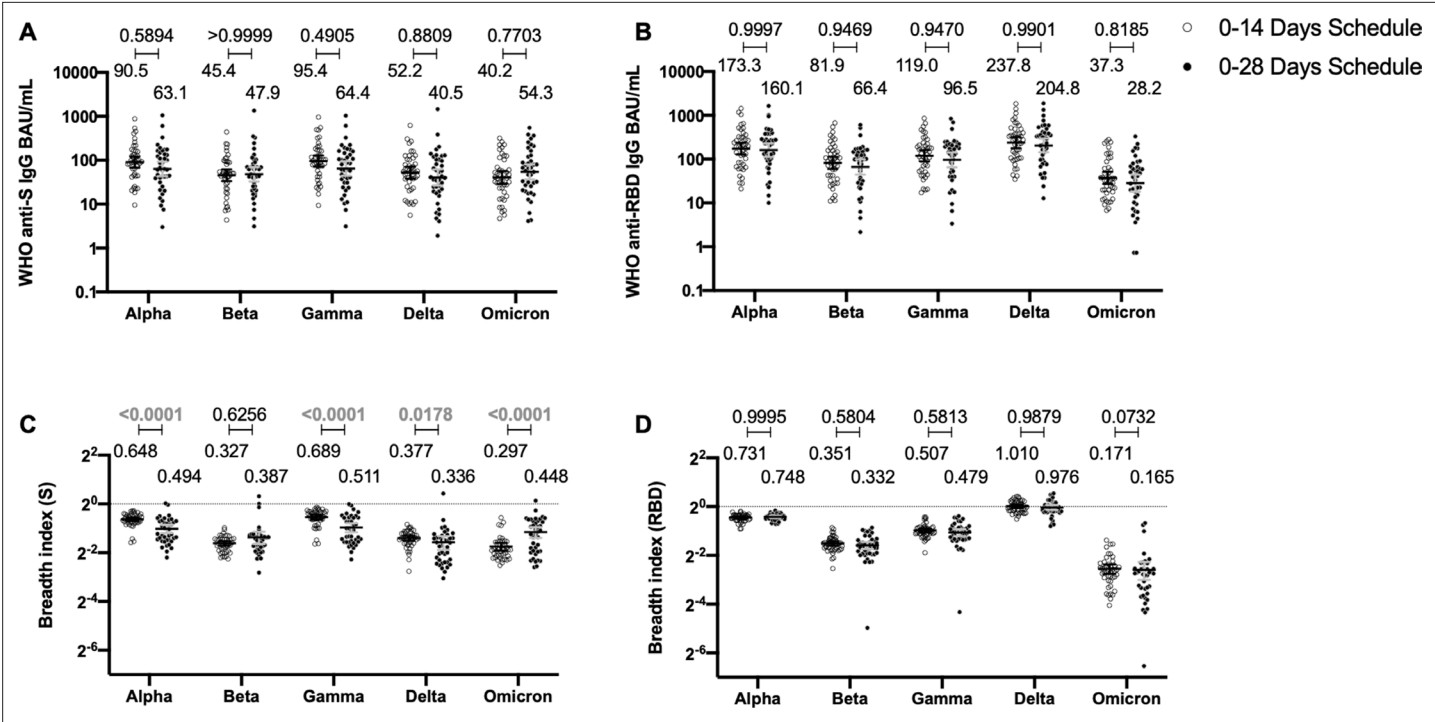

**Figure 4.** Antibodies against the spike (S) and receptor-binding domain (RBD) from variant of concern (VOC) of severe acute respiratory syndrome coronavirus 2 (SARS-CoV-2) are similar between schedules, while breadth index varies between schedules. Antibodies concentrations against the S (**A**) and the RBD (**B**) of different VOCs of SARS were evaluated through meso-scale discovery (MSD). n=44 volunteers for the 0–14 schedule. n=40 volunteers for the 0–28 schedule. Samples evaluated were obtained at 4 weeks after the second dose. Data is represented as the reciprocal antibody titer of neutralizing antibody versus the different VOCs evaluated. With these values, a breadth index was calculated for each VOC for anti-S (**C**) and anti-RBD (**D**) antibodies. Numbers above the bars show either the international units (IU) (**A, B**) or the breadth index (**C, D**), and the error bars indicate the 95% CI. Data were analyzed by a mixed-effect two-way ANOVA, followed by a Bonferroni's post hoc test to compare immunization schedules. Numbers above each bracket represent calculated p values comparing both immunization schedules. Statistical significance was set at p<0.05 and highlighted numbers indicate statistical significance.

The online version of this article includes the following source data and figure supplement(s) for figure 4:

**Source data 1.** Data used to generate *Figure 4* and *Figure 4—figure supplements 1–3*.

**Figure supplement 1.** Antibodies against the spike (S) and receptor-binding domain (RBD) from variant of concern (VOC) of severe acute respiratory syndrome coronavirus 2 (SARS-CoV-2) and their breadth indexes are reduced relative to the Ancestral strain, except in the receptor-binding domain (RBD)-related parameters for the Delta strain.

**Figure supplement 2.** Antibodies against the spike (S) and receptor-binding domain (RBD) from variant of concern (VOC) of severe acute respiratory syndrome coronavirus 2 (SARS-CoV-2) are similar between schedules, while breadth index varies between schedules for the 18–59 years age group.

**Figure supplement 3.** Antibodies against the spike (S) from variant of concern (VOC) of severe acute respiratory syndrome coronavirus 2 (SARS-CoV-2) and breadth indexes vary between schedules for the >60 years age group, while these parameters for anti-receptor-binding domain (RBD) antibodies remain similar.

**Figure supplement 4.** Neutralizing antibodies against the receptor-binding domain (RBD) from variant of concern (VOC) of severe acute respiratory syndrome coronavirus 2 (SARS-CoV-2) are similar between schedules, while breadth index varies between schedules.

on T cells (*Figure 3—figure supplement 2*). No differences could be found between both immunization schedules for the number of IL-4$^+$ SFCs (*Figure 3—figure supplement 3*). Overall, these results indicate that immunization with CoronaVac in both schedules induces an increase in the number of IFN-γ$^+$ SFC and the expression of AIM by CD4$^+$ and CD8$^+$ T cells upon stimulation with several MPs.

## Immunization with CoronaVac induces a similar profile of antibodies against the S protein and the RBD of SARS-CoV-2 VOC regardless of the immunization schedule

To determine whether the immunization schedule had any impact on the profile of antibodies elicited against SARS-CoV-2 VOCs, MSD immunoassays were performed to determine antibody titers against the S protein or the RBD from SARS-CoV-2 VOC (*Figure 4*). Samples from 44 volunteers in the 0–14 schedule and from 40 volunteers in the 0–28 schedule, obtained 4 weeks after the second dose, were evaluated. No differences were seen for the titers of anti-S and anti-RBD antibodies between the two schedules, when both age groups were evaluated together (*Figure 4A, B*). Anti-S antibodies against all variants tested (Alpha, Beta, Gamma, Delta, and Omicron) showed decreased concentrations compared to the Ancestral strain regardless of the immunization schedule (*Figure 4—figure supplement 1A*). A similar trend was observed for anti-RBD antibodies, although the concentration of antibodies against the Delta strain RBD seemed to remain similar to antibody levels against the Ancestral strain RBD (*Figure 4—figure supplement 1B*). The concentration of antibodies that recognize the Omicron RBD seemed to decrease in a more pronounced way compared to the decrease observed for antibodies that recognize the Omicron S protein. When both age groups were analyzed independently, no differences were seen in the 18–59 years age group for anti-S and anti-RBD antibodies (*Figure 4—figure supplement 2A, B*). For the >60 years age group, decreased anti-S antibodies concentrations were found against the Alpha, Gamma, and Delta in the 0–28 schedule, compared to the 0–14 schedule (*Figure 4—figure supplement 3A*). No differences in antibodies concentrations against the RBD were found for this age group, irrespective of the schedule (*Figure 4—figure supplement 3B*).

To account for differences on the antibody-binding activity of each volunteer's serum, we calculated a breadth index, defined as the concentration of antibodies for a particular VOC divided by the concentration of antibodies for the Ancestral strain. Differences in breadth of antibodies were found for the anti-S antibodies, with the 0–28 schedule showing decreased recognition capacity for Alpha, Gamma, and Delta, relative to the 0–14 schedule (*Figure 4C*). Interestingly, the 0–28 schedule showed increased recognition of the Omicron VOC, compared to the 0–14 schedule. We found no differences in this index for anti-RBD antibodies between schedules (*Figure 4D*). A decreased recognition capacity for all VOC of anti-S antibodies was detected (*Figure 4—figure supplement 1C*). This was also seen in most VOC for anti-RBD antibodies, but no differences were found for the Delta strain (*Figure 4—figure supplement 1D*). When analyzing the age groups, the 18–59 years age group exhibited a reduced breadth index for the Alpha VOC in the 0–28 schedule, compared to the 0–14 schedule (*Figure 4—figure supplement 2C*). No differences were seen for RBD (*Figure 4—figure supplement 2D*). For the >60 years age group, the 0–28 schedule exhibited a reduced breadth index for the Alpha, Gamma, and Delta VOC, compared to the 0–14 schedule (*Figure 4—figure supplement 3C and D*). Remarkably, the 0–28 schedule reported increased breadth index for the Beta and Omicron VOC, relative to the 0–14 schedule (*Figure 4—figure supplement 3C and D*).

Finally, to further support our data, we evaluated the neutralization capacity against these different VOCs using the cVNT technique described in *Figure 1*. *Figure 4—figure supplement 4* shows no differences in neutralization against VOCs between both immunization schedules, regardless of the technique performed, except for the Gamma variant evaluated, where the 0–14 schedule showed higher values than the 0–28 schedule (*Figure 4—figure supplement 4*). Interestingly, the 0–28 schedule exhibited a reduced breadth index for the Gamma and Omicron VOCs.

These results suggest that CoronaVac may induce moderate cross-reactive humoral immune responses against SARS-CoV-2 VOC. While no evident patterns could be found between both schedules, the most marked differences can be detected when evaluating the >60 years age group.

## The humoral and cellular immune responses elicited by both immunization schedules exhibit a significant correlation pattern

To identify potential correlations between variables, we generated correlation matrixes for each immunization schedule (*Figure 5A and D*). Overall, the neutralizing capacities determined by the different techniques exhibited a positive correlation value for both schedules. Particularly, the neutralizing capacities of circulating antibodies were strongly and positively correlated for the cVNT and sVNT evaluation at 4 weeks after the second dose (*Figure 5B and E*). These correlations were statistically

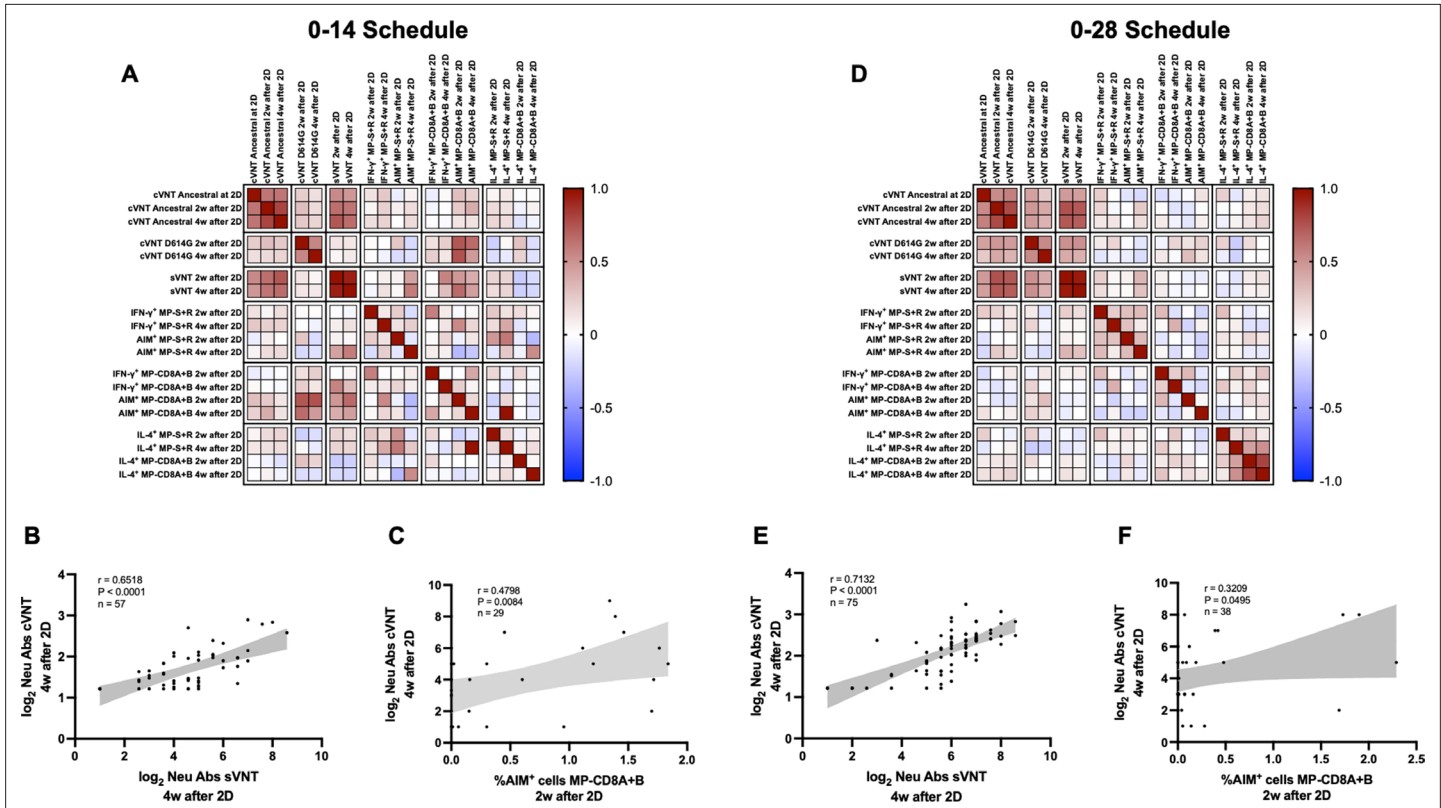

**Figure 5.** Multivariate analyses show correlated humoral and cellular immune responses. Pearson correlation matrixes were generated independently for the 0–14 (**A**) and 0–28 (**D**) immunization schedules, including humoral and cellular immune response variables. Colors indicate r values, and the scale is shown next to each matrix. Individual selected Pearson correlations for the 0–14 (**B–C**) and 0–28 (**E–F**) immunization schedules are shown, indicating n, r, and p values. Statistical significance was set at p<0.05. Shaded gray areas show the 95% CI of the correlations.

significant and were found for both immunization schedules. Interestingly, positive correlations were also found for the neutralizing capacities of circulating antibodies as determined by cVNT 2 weeks after the second dose and the expression of AIM by CD8[+] T cells 4 weeks after the second dose (*Figure 5C and F*). Again, these correlations were statistically significant and were found for both immunization schedules. These results suggest that the immune responses elicited by CoronaVac go hand in hand for either immunization schedule, as increased values of neutralizing antibodies are associated with increased expression of AIM by T cells.

## Discussion

Both SARS-CoV-2 and the ongoing COVID-19 pandemic have taken a considerable toll on the population worldwide and have posed a significant burden on the social well-being and the economies of developing and developed countries (*Dong et al., 2020*). Scientific efforts have been directed toward generating safe and effective vaccines to prevent disease and severe cases associated with this virus, and the strive has not been vain (*World Health Organization, 2022*). Here, we report the immunogenicity profile elicited by healthy adults enrolled in a multicenter, randomized, and controlled phase 3 clinical trial performed with CoronaVac, a whole virus-inactivated SARS-CoV-2 vaccine manufactured by Sinovac Life Sciences, in Chile (*Gao et al., 2020*). A total of 2302 volunteers were enrolled in this phase 3 trial, and a subset of these volunteers had their immunogenicity parameters evaluated. Circulating antibodies exhibited enhanced neutralizing capacities against SARS-CoV-2, as determined by four different assays. Differential responses antibodies concentrations against the S and the RBD of SARS-CoV-2 and different VOCs were detected, with breadth indexes indicating varying degrees of changes against these strains. Increased secretion of IFN-γ and expression of AIM were detected in T cells upon stimulation of PBMCs with MPs from SARS-CoV-2, as measured by ELISPOT and

flow cytometry assays, respectively. We also performed multivariate correlation analyses to evaluate possible correlations between the measured parameters.

A prospective study considering a cohort of >10 million vaccinated persons in a 0–28 immunization schedule showed that the effectiveness of CoronaVac for this trial is 65.9% for preventing the development of COVID-19 symptoms, while a 90.3% was reported for the prevention of intensive care unit (ICU) admission (*Jara et al., 2021*). Studies from phase 1/2 clinical trials in China suggested that a 0–28 schedule may better induce a protective response against SARS-CoV-2 with CoronaVac (*Bueno et al., 2022*; *Wu et al., 2021*; *Zhang et al., 2021*). Therefore, here we compared the immune response elicited by both schedules. Recent reports show that antibody titers against SARS-CoV-2 correlate with the protective capacities of COVID-19 vaccines (*Earle et al., 2021*). Immunization with CoronaVac induced significantly increased levels of circulating neutralizing antibodies for both immunization schedules at all times after the first and second dose, irrespective of the age group evaluated. This is an expected response for vaccines that promote a protective response against SARS-CoV-2 and has also been reported for other vaccines such as BNT162b2 and mRNA-1273 (*Tarke et al., 2021*).

The previous phase 1/2 trials held in China showed that a 0–28 immunization schedule is better at inducing anti-S1 and neutralizing antibodies in healthy adults, while in children and adolescents, it causes higher seroconversion rates before the second dose, compared to the 0–14 schedule (*Han et al., 2021*; *Zhang et al., 2021*). Here, we show similar data regarding the neutralizing capacities of circulating antibodies, as arbitrary IU and GMT values were mainly increased 2 and 4 weeks after the second dose for the 0–28 schedule compared to the 0–14 schedule. It is important to note that the inactivated virus used in CoronaVac is the strain CZ02 (*Gao et al., 2020*), while the cVNT, sVNT, and pVNT use either the circulating Ancestral and D614G strain, the S1-RBD, or the S protein from the original wild-type L strain (*Beltrán-Pavez et al., 2021*), respectively. This could explain differences in neutralizing capacities reported among these techniques. Remarkably, no differences were found between schedules for the concentration of antibodies against the S protein and the RBD, at 4 weeks after the second dose. However, reduced breadth indexes were seen for the different VOC evaluated, with increased values against Beta and Omicron, but decreased values against the other variants in the 0–28 schedule, suggesting that a booster dose may be required to promote a more robust protective humoral immune responses against these emerging viruses (*Schultz et al., 2022*). Remarkably, as seen in *Figure 2*, although similar levels of antibodies against the S and the RBD of the Ancestral strain of SARS-CoV-2 were found, there were still differences in the neutralization capacities of these antibodies for each schedule, as seen in *Figure 1*. These differences may be significant if assuming that the presence of total antibodies is protective against an infection, as suggested for other respiratory viruses (*Polack et al., 2003*; *Polack et al., 2002*; *Winarski et al., 2019*). However, the use of rapid tests for the evaluation of antibodies in humans should not be used as a proxy of actual protection against disease since neutralization will be overlooked in this context.

We also performed assays to evaluate the neutralizing capacity of antibodies against VOCs. Overall, we did not observe significant differences in the neutralizing antibody titers against VOCs when comparing both immunization schedules, except for the Gamma strain, where the 0–14 schedule showed higher values than those of the 0–28 schedule. Interestingly, the breath indexes determined for these assays suggests an enhanced affinity across VOCs in the 0–14 schedule. Further studies that address the spatial antibody structure elicited in both vaccination schedules will be needed to understand the structural bases that could explain these observed differences, as the RBD-antibody interface affects the breadth of each antibody to viral escape (*Starr et al., 2021*). Finally, these results should also be interpreted with caution, as larger sample sizes for each vaccination schedule are needed for a more definitive conclusion.

Remarkably, it was recently shown that two doses of CoronaVac induce a steady cellular immune response against circulating VOCs of SARS-CoV-2, while the neutralizing capacities of circulating antibodies were different among different strains (*Melo-González et al., 2021*). This robust cellular response may be related to the presence of viral antigens other than the S protein of this virus and could be key when choosing vaccines to face this pandemic. The T cell immune responses elicited upon natural infection and vaccination are fundamental in modulating the disease caused by SARS-CoV-2 (*Canedo-Marroquín et al., 2020*; *Duarte et al., 2021*). Accordingly, several of the currently WHO-approved vaccine platforms have been shown to induce potent cellular immune responses, including those composed of recombinant proteins, mRNA, and viral vectors (*Ewer et al., 2021*;

*Keech et al., 2020*). As reported previously, MP-S and MP-R were generated in silico to stimulate mainly CD4$^+$ T cells (i.e., peptides with 15-mer length in these MPs), while MP-CD8A and MP-CD8B were generated to stimulate mainly CD8$^+$ T cells (i.e., peptides with 9- to 11-mer length in these MPs) (*Grifoni et al., 2020*). Since both T cell responses are relevant during a viral infection, both sets of MPs were tested to evaluate the elicited immune response. The expression of AIM by T cells was mostly similar for both schedules at 2 and 4 weeks after the second dose, with no statistical differences. However, stimulation with MPs of 15-mer peptides induced an increased expression of AIM by CD4$^+$ T cells relative to preimmune samples, 2 and 4 weeks after the second dose. The responses measured here indicate that a cellular response can be detected upon stimulation with these MPs initially selected in silico (*Grifoni et al., 2020*). A Th1 response commonly associated with IFN-γ secretion is optimal for the clearance of intracellular pathogens, while Th2-related cytokines such as IL-4 may inhibit the polarization of CD4$^+$ T cells toward this antiviral profile (*Bueno et al., 2022*; *Wu et al., 2021*; *Zhang et al., 2021*). Although statistically higher than those detected for preimmune samples when stimulating with MPs of 15-mer peptides, numbers of IL-4$^+$ SFC were remarkably lower than those seen for IFN-γ$^+$ SFC. This is in line with the data previously reported for the 0–14 immunization schedule (*Bueno et al., 2022*).

Our correlation matrixes showed that both immunization schedules could promote concerted humoral and cellular immune responses. Specifically, circulating neutralizing antibodies measured by different techniques were highly correlated 2 and 4 weeks after administration of the second dose. Positive and statistically significant correlations were found between neutralizing antibody titers determined by cVNT against the D614G variant 2 and 4 weeks after the second dose and the expression of AIM by CD8$^+$ T cells 4 weeks after the second dose for both immunization schedules. This is especially important as both humoral and cellular immunity contribute to viral clearance, and vaccines should aim to develop both arms of the adaptive immunity (*Choudhary et al., 2021*).

This study also has caveats and limitations. Although the robust immunogenicity described here is encouraging, efficacy, hospitalization, and death prevention analysis are required to guide the use of this vaccine (*Wu et al., 2021*; *Zhang et al., 2021*). Other limitations of this study are the lack of evaluation of long-term immunity (i.e., 6 or 12 months after the first dose), the partially homogeneous ethnicity of the evaluated population (healthy adults), a more exhaustive evaluation of cellular responses, and the immune response elicited against circulating variants of this virus.

Considering all the data presented in this article, we can conclude that immunization with CoronaVac in either a 0–14 or 0–28 vaccination schedule induces robust humoral and cellular responses in healthy adults from Chile. Further studies related to this phase 3 trial will be focused on the response elicited at later times after vaccination (i.e., 6 and 12 months after the first dose), the protection of CoronaVac toward circulating SARS-CoV-2 variants, and the capacity of a third dose to induce a robust immune response.

## Acknowledgements

We would like to thank the Ministry of Health, Government of Chile; Ministry of Science, Technology, Knowledge, and Innovation, Government of Chile; The Ministry of Foreign Affairs, Government of Chile; the Chilean Public Health Institute (ISP); and The Confederation of Production and Commerce (CPC), Chile. We also would like to thank PATH for their support and sharing the First WHO International Standard for anti-SARS-CoV-2 immunoglobulin. We are grateful to Rami Scharf, Jessica White, Jorge Flores, and Miren Iturriza-Gomara from PATH for their support on experimental design and discussion; Alex Cabrera and Sergio Bustos from the Flow Cytometry Facility at Facultad de Ciencias Biológicas, Pontificia Universidad Católica de Chile for support with flow cytometry. We also thank the Vice Presidency of Research (VRI), the Direction of Technology Transfer and Development (DTD), the Legal Affairs Department (DAJ) of the Pontificia Universidad Católica de Chile. We are also grateful to the Administrative Directions of the School of Biological Sciences and the School of Medicine of the Pontificia Universidad Católica de Chile for their administrative support. Special thanks to the independent data safety monitoring committee (members in the Supplementary Appendix [SA]) for their oversight and to the subjects enrolled in the study for their participation and commitment to this trial. Members of the CoronaVac03CL Study Team are listed in the SA.

## Additional information

### Competing interests

Ricardo Soto-Rifo: has received funding from ANID - ICM, ICN 2021_045. The author has no other competing interests to declare. Daniela Weiskopf: has received funding support from the NIH under contract number 75N93019C00065. The La Jolla Institute for Immunology (LJI) has filed for patent protection for various aspects of T cell epitope and vaccine design work. The author has no other competing interests to declare. Alba Grifoni: The La Jolla Institute for Immunology (LJI) has filed for patent protection for various aspects of T cell epitope and vaccine design work. The author has no other competing interests to declare. Alessandro Sette: is a consultant for Gritstone Bio, Flow Pharma, Arcturus, Immunoscape, CellCarta, Moderna, AstraZeneca, Fortress, Repertoire, Gilead, Gerson Lehrman Group, RiverVest, MedaCorp, Guggenheim, OxfordImmunotech, and Avalia. The author has received funding support from the NIH under contract 75N93021C00016 and 75N93019C00065. The La Jolla Institute for Immunology (LJI) has filed for patent protection for various aspects of T cell epitope and vaccine design work. The author has no other competing interests to declare. Gang Zeng, Weining Meng: is a SINOVAC employee and contributed to the conceptualization of the study (clinical protocol and eCRF design). CoronaVacCL03 Study Group: Pablo A González: acts as the Executive Director of the clinical trials PedCoronaVac03CL clinical study (ClinicalTrials.gov NCT04992260) and CoronaVac03CL (ClinicalTrials.govNCT04651790) (funds to the institution), and receives research support from Millennium Institute on Immunology and Immunotherapy. The author received funding from Agencia Nacional de Investigación y Desarrollo, Fondo de Fomento al Desarrollo Científico y tecnológico. The author has no other competing interests to declare. Susan M Bueno: acts as the Scientific Director of clinical trials PedCoronaVac03CL clinical study (ClinicalTrials.gov NCT04992260) and CoronaVac03CL (ClinicalTrials.govNCT04651790) (funds to the institution), and receives research support from Millennium Institute on Immunology and Immunotherapy. The author has received funding from Agencia Nacional de Invetsigación y Desarrollo, Fondo de Fomento al Desarrollo Científico y tecnológico ID20I10082. The author has no other competing interests to declare. Alexis M Kalergis: acts as the General Director of clinical trials PedCoronaVac03CL clinical study (ClinicalTrials.gov NCT04992260) and CoronaVac03CL (ClinicalTrials.govNCT04651790). The author has received funding from Agencia Nacional de Investigació n y Desarrollo (ANID) - Millennium Science Initiative Program - ICN09_016 / ICN 2021_045: Millennium Institute on Immunology and Immunotherapy (ICN09_016 / ICN 2021_045; former P09/016-F) and Agencia Nacional de Investigación y Desarrollo [FONDECYT grant numbers 1190830]. The author has no other competing interests to declare. The other authors declare that no competing interests exist.

### Funding

| Funder | Grant reference number | Author |
| --- | --- | --- |
| The Ministry of Health, Government of Chile | | CoronaVacCL03 Study Group |
| The Confederation of Production and Commerce (CPC), Chile | | CoronaVacCL03 Study Group |
| The Millennium Institute on Immunology and Immunotherapy, ANID - Millennium Science Initiative Program ICN09_016 (former P09/016-F) | | Pablo A González |
| The Innovation Fund for Competitiveness FIC-R 2017 (BIP Code: 30488811-0) | | Pablo A González |
| FONDECYT grant | 1190156 | Ricardo Soto-Rifo |

| Funder | Grant reference number | Author |
| --- | --- | --- |
| FONDECYT grant | 1180798 | Fernando Valiente-Echeverría |
| NIH NIAID Contract | 75N93021C00016 | Alessandro Sette |
| NIH NIAID Contract | 75N9301900065 | Daniela Weiskopf |
| Bill & Melinda Gates Foundation | INV- 021239 and INV-016821 | Pablo A González |
| PATH | UC_GAT.583854-01709560-COL | Alexis M Kalergis |

The funders had no role in study design, data collection and interpretation, or the decision to submit the work for publication.

## Author contributions

Nicolás MS Gálvez, Gaspar A Pacheco, Conceptualization, Data curation, Formal analysis, Validation, Investigation, Methodology, Writing – original draft, Writing – review and editing; Bárbara M Schultz, Formal analysis, Validation, Investigation, Visualization, Methodology, Writing – review and editing; Felipe Melo-González, Liliana A González, Formal analysis, Validation, Investigation, Methodology, Writing – review and editing; Jorge A Soto, Luisa F Duarte, Validation, Investigation, Methodology, Writing – review and editing; Daniela Rivera-Pérez, Mariana Ríos, Roslye V Berrios, Yaneisi Vázquez, Daniela Moreno-Tapia, Omar P Vallejos, Catalina A Andrade, Guillermo Hoppe-Elsholz, Carolina Iturriaga, Álvaro Rojas, Investigation, Methodology, Writing – review and editing; Marcela Urzua, María S Navarrete, Rodrigo Fasce, Jorge Fernández, Judith Mora, Eugenio Ramírez, Aracelly Gaete-Argel, Mónica L Acevedo, Fernando Valiente-Echeverría, Ricardo Soto-Rifo, Methodology, Writing – review and editing; Daniela Weiskopf, Alba Grifoni, Alessandro Sette, David Goldblatt, Resources, Methodology, Writing – review and editing; Gang Zeng, Weining Meng, Conceptualization, Resources, Methodology, Writing – review and editing; CoronaVacCL03 Study Group, Investigation; José V González-Aramundiz, Conceptualization, Visualization, Methodology, Writing – review and editing; Marina Johnson, Investigation, Visualization, Methodology; Pablo A González, Alexis M Kalergis, Conceptualization, Funding acquisition, Methodology, Project administration, Writing – review and editing; Katia Abarca, Conceptualization, Supervision, Project administration, Writing – review and editing; Susan M Bueno, Conceptualization, Supervision, Funding acquisition, Validation, Methodology, Project administration, Writing – review and editing

## Author ORCIDs

Nicolás MS Gálvez ⓘ http://orcid.org/0000-0003-2046-7882
Gaspar A Pacheco ⓘ http://orcid.org/0000-0001-5748-5027
Bárbara M Schultz ⓘ http://orcid.org/0000-0002-6201-6978
Felipe Melo-González ⓘ http://orcid.org/0000-0002-3711-2407
Jorge A Soto ⓘ http://orcid.org/0000-0003-0335-9759
Luisa F Duarte ⓘ http://orcid.org/0000-0003-1828-1800
Catalina A Andrade ⓘ http://orcid.org/0000-0003-2623-1387
Pablo A González ⓘ http://orcid.org/0000-0001-7709-6870
Katia Abarca ⓘ http://orcid.org/0000-0003-0404-3887
Susan M Bueno ⓘ http://orcid.org/0000-0002-7551-8088
Alexis M Kalergis ⓘ http://orcid.org/0000-0001-7622-5263

## Ethics

Clinical trial registration NCT04651790.
This clinical trial (clinicaltrials.gov NCT04651790) is a randomized and controlled study held in Chile with eight different sites. The study protocol adhered to the current Tripartite Guidelines for Good Clinical Practices, the Declaration of Helsinki, and local regulations and was approved by the Institutional Scientific Ethical Committee of Health Sciences of the Pontificia Universidad Católica de Chile, (#200708006). The execution was approved by the Chilean Public Health Institute (#24204/20).

## Decision letter and Author response

Decision letter https://doi.org/10.7554/eLife.81477.sa1

Author response https://doi.org/10.7554/eLife.81477.sa2

## Additional files

### Supplementary files
• MDAR checklist

### Data availability
All raw data (anonymized to protect the information of volunteers) is included with the publication of this article as a supporting file. Each Source Data File contain the numerical data used to generate all the figures. The study protocol is also available online and was previously published in https://doi.org/10.1101/2021.03.31.21254494.

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
