## [Editor Report]

This manuscript investigates the humoral neutralizing antibody and cellular immune responses of volunteers in a randomized clinical trial for the CoronaVac SARS-CoV-2 vaccine. The findings are useful and provide context for the efficaciousness of the 0-14 day and 0-28 day dosing schedules of CoronaVac. The results show that these two dosing schedules are similar across most metrics.

---

## [Decision Letter]

**Decision letter after peer review:**

Thank you for submitting your article "Differential immune response induced by two immunization schedules with an inactivated SARS-CoV-2 vaccine in a randomized phase 3 clinical trial" for consideration by *eLife*. Your article has been reviewed by 3 peer reviewers, one of whom is a member of our Board of Reviewing Editors, and the evaluation has been overseen by Mone Zaidi as the Senior Editor. The reviewers have opted to remain anonymous.

Essential revisions:

1) Please provide data for the emerging variants of concern with at least one of the three neutralization antibody assays that the authors use for the 2020 strain in Figure 1. This will help identify the most important differences between the two immunization schedules with regards to the immunization schedule which I feel is somewhat unclear from the data on overall anti-Spike and anti-RBD antibodies.

2) Please correct the issues identified by reviewers 1, 2, and 3.

*Reviewer #1 (Recommendations for the authors):*

Overall this is a nice manuscript with ample data. There are some typos (such as point one below). The major weakness in significance was the lack of a mRNA comparison arm to understand the magnitude of the responses and dosing schedule. That likely cannot be overcome, However, the lack of long-term follow up as noted in the manuscript is likely something that would be important to substantiate our premise that a booster would be needed or of value and at what time.

1) In figure 1C, the 2w after the 2nd dose has the wrong numbers (89 and 89).

2) In figure 2, there is a fairly conclusive demonstration that there are relatively similar total antibody quantitative levels between the two dosing regimes, but this is in contrast to the neutralization data. This has important clinical implications in that these more rapid and easy assays cannot be used as a proxy for protection as gauged by neutralization.

*Reviewer #2 (Recommendations for the authors):*

My major recommendation to improve this manuscript would be to provide data for the emerging variants of concern with at least one of the three neutralization antibody assays that the authors use for the 2020 strain in Figure 1. This will help identify the most important differences between the two immunization schedules with regards to the immunization schedule which I feel is somewhat unclear from the data on overall anti-Spike and anti-RBD antibodies. In addition, I have the following concerns:

1. Title of the manuscript: the overall message appears to be that both immunization schedules provide broadly similar protection except for the neutralization antibody titre. However, the title would suggest that the differences are more specific. Tying into my point on the usefulness of the data for neutralization across VOC, the additional data, if provided, could sharpen the message of the paper around the neutralizing antibody capacity.

2. Jargon: The text introduces a lot of jargon directly in the abstract and early in the introduction without any mention of what the term is or what it means. Examples include " MPs, sVNT, cVNT, pVNT, SFC, AIM, PBMCs". In many instances, these terms are adequately defined in the legends of Figures 1 to 3. For ease of reading and for a more general audience, could the authors please move these definitions to the main text?

3. Order of presentation of results: For Figures 3 and S4, S5, as well as Figures 4 and S9; the text presents the data in the supplementary figures first before discussing the main figures. The supplementary figures show pairwise comparisons between the pre-immune and post-vaccination states, whereas the main text provides the comparison between each immunization state. While I agree that the latter is more interesting, I think the presentation could be much simplified by including all the comparisons directly in the main figures. That way, the obvious point (i.e., the vaccine works) can be more efficient compared with the point of interest to the authors (i.e., the differences between the schedules).

4. Data on MPs: I am somewhat confused why the second set of MPs (CD8A and CD8B), which the text identifies as two parts of the whole proteome of SARS-CoV-2 gave different results from MP-R which also contains the non-spike components of the proteome. Is this because of the difference in length of the peptides? It would help if the authors clarified this point as, at the moment, I do not quite see what this assay brings to the manuscript.

*Reviewer #3 (Recommendations for the authors):*

This study is conducted well and written well. Additional understanding underlying the reported efficaciousness of COVID vaccines, especially against variants of concern, is always welcome. My comments are below:

1. The text within the figure plots (P-values, etc.) is quite small and hard to read. Given the figure format of *eLife*, can the authors please increase the size of such text? The descriptive statistics bars (mean +/-) are also very difficult to see.

2. If the results are to be placed before the methods section, then many of the acronyms should be specified when they are first mentioned in the Results section (sVNT, cVNT, etc.)

3. In the methods section, the authors mention, "Inclusion and exclusion criteria were reported previously [13]". The authors briefly describe the exclusion criteria following this sentence. Can the authors briefly describe the inclusion criteria to help with reader comprehension of this study?

4. The tables are missing units (e.g., days, years).

5. The age-related effects of SARS-CoV-2 vaccination are an area of high interest, scientifically and clinically. Given that this study includes such data, can the authors include statistical comparisons on how age influenced circulating neutralizing antibody quantitation?

6. The results in the Breadth index (S) between 0-14 and 0-28 schedules are interesting; however, this conclusion on potential cross-reactive immunity must be made carefully. Given that the S compares the Ab conc for a VOC with that of the original strain, the authors should conduct a paired sample analysis instead.

---

## [Author Response]

Essential revisions:(1) Please provide data for the emerging variants of concern with at least one of the three neutralization antibody assays that the authors use for the 2020 strain in Figure 1. This will help identify the most important differences between the two immunization schedules with regards to the immunization schedule which I feel is somewhat unclear from the data on overall anti-Spike and anti-RBD antibodies.

As requested by the Editor and Reviewer, we have included in the revised version of the manuscript new data regarding the neutralization of circulating variants of concern of SARS-CoV-2 using the cVNT technique (Figure 4—figure supplement 4).

(2) Please correct the issues identified by reviewers 1, 2, and 3.

As requested by the Editor, we have addressed all the comments made by Reviewers 1, 2, and 3, as indicated in the Point-by-Point response below, and modified the respective manuscript sections.

We would like to kindly thank the Reviewers and Editors for their time and effort in the handling this manuscript and expect that the current revised version is suitable for publication in *eLife*.

Reviewer #1 (Recommendations for the authors):Overall this is a nice manuscript with ample data. There are some typos (such as point one below). The major weakness in significance was the lack of a mRNA comparison arm to understand the magnitude of the responses and dosing schedule. That likely cannot be overcome, However, the lack of long-term follow up as noted in the manuscript is likely something that would be important to substantiate our premise that a booster would be needed or of value and at what time.

We would like to thank the Reviewer for the positive comments. We have addressed all the concerns and comments raised in the following points.

(1) In figure 1C, the 2w after the 2nd dose has the wrong numbers (89 and 89).

Thank you for noticing this. As requested, we have updated the figure to fix this issue.

(2) In figure 2, there is a fairly conclusive demonstration that there are relatively similar total antibody quantitative levels between the two dosing regimes, but this is in contrast to the neutralization data. This has important clinical implications in that these more rapid and easy assays cannot be used as a proxy for protection as gauged by neutralization.

As suggested by the Reviewer, we have updated the manuscript to comment on this regard in the Discussion section (Lines 303-310).

Reviewer #2 (Recommendations for the authors):My major recommendation to improve this manuscript would be to provide data for the emerging variants of concern with at least one of the three neutralization antibody assays that the authors use for the 2020 strain in Figure 1. This will help identify the most important differences between the two immunization schedules with regards to the immunization schedule which I feel is somewhat unclear from the data on overall anti-Spike and anti-RBD antibodies. In addition, I have the following concerns:

As requested by the Reviewer, in the revised version of the manuscript we have added new data and information regarding the neutralization of variants of concern using the cVNT methodology. We agree with the Reviewer in that this provides to the reader further insights regarding the differences between the vaccination schedules (Figure 4—figure supplement 4, Lines 226-231 and 311-321).

1. Title of the manuscript: the overall message appears to be that both immunization schedules provide broadly similar protection except for the neutralization antibody titre. However, the title would suggest that the differences are more specific. Tying into my point on the usefulness of the data for neutralization across VOC, the additional data, if provided, could sharpen the message of the paper around the neutralizing antibody capacity.

As suggested by the Reviewer, we have included in the revised version of the manuscript additional information regarding the neutralization of variants of concern upon the two vaccination schedules and modified the title of the manuscript accordingly (Figure 4—figure supplement 4, Lines 2, 226-231, and 311-321).

2. Jargon: The text introduces a lot of jargon directly in the abstract and early in the introduction without any mention of what the term is or what it means. Examples include " MPs, sVNT, cVNT, pVNT, SFC, AIM, PBMCs". In many instances, these terms are adequately defined in the legends of Figures 1 to 3. For ease of reading and for a more general audience, could the authors please move these definitions to the main text?

As requested by the Reviewer, we have relocated the definitions of the abbreviated terms indicated in the manuscript to the main text, such as in the introduction or Results sections.

3. Order of presentation of results: For Figures 3 and S4, S5, as well as Figures 4 and S9; the text presents the data in the supplementary figures first before discussing the main figures. The supplementary figures show pairwise comparisons between the pre-immune and post-vaccination states, whereas the main text provides the comparison between each immunization state. While I agree that the latter is more interesting, I think the presentation could be much simplified by including all the comparisons directly in the main figures. That way, the obvious point (i.e., the vaccine works) can be more efficient compared with the point of interest to the authors (i.e., the differences between the schedules).

As requested by the Reviewer, we have modified Figure 3 to show all the statistical differences between groups. Consequently, we have removed Figures S4 and S5 from the manuscript. Nevertheless, we have kept Figure 4—figure supplement 1 (previously Figure S9) to adequately address a comment raised by another Reviewer, aimed at evaluating the differences with the corresponding variants of concern more adequately, while rearranging the text to cite Figure 4—figure supplement 1 after Figure 4 (Lines 199-219).

4. Data on MPs: I am somewhat confused why the second set of MPs (CD8A and CD8B), which the text identifies as two parts of the whole proteome of SARS-CoV-2 gave different results from MP-R which also contains the non-spike components of the proteome. Is this because of the difference in length of the peptides? It would help if the authors clarified this point as, at the moment, I do not quite see what this assay brings to the manuscript.

Thank you for this comment. As requested by the Reviewer, we have modified the text to discuss in more detail these results in the Discussion section. Comments were added regarding the differences between these MPs and potential CD4^+^ and CD8^+^ T cell responses (Lines 332-337).

Reviewer #3 (Recommendations for the authors):This study is conducted well and written well. Additional understanding underlying the reported efficaciousness of COVID vaccines, especially against variants of concern, is always welcome. My comments are below:1. The text within the figure plots (P-values, etc.) is quite small and hard to read. Given the figure format of eLife, can the authors please increase the size of such text? The descriptive statistics bars (mean +/-) are also very difficult to see.

As requested by the Reviewer, we have modified the text in the figures so that they are easier to read.

2. If the results are to be placed before the methods section, then many of the acronyms should be specified when they are first mentioned in the Results section (sVNT, cVNT, etc.)

As requested by the Reviewer, we have relocated the definition of these acronyms upon their first appearance in the manuscript so that the reader can access them more easily (Introduction and Results sections).

3. In the methods section, the authors mention, "Inclusion and exclusion criteria were reported previously [13]". The authors briefly describe the exclusion criteria following this sentence. Can the authors briefly describe the inclusion criteria to help with reader comprehension of this study?

As requested by the Reviewer, we have included a brief description of these criteria in the text (Lines 386-391).

4. The tables are missing units (e.g., days, years).

As requested by the Reviewer, we have updated the tables accordingly to add the missing units.

5. The age-related effects of SARS-CoV-2 vaccination are an area of high interest, scientifically and clinically. Given that this study includes such data, can the authors include statistical comparisons on how age influenced circulating neutralizing antibody quantitation?

As requested by the Reviewer, we have updated Table 1 and Figure 1—figure supplement 3 to indicate comparisons for the neutralization data based on the age of the individuals in each vaccination schedule.

6. The results in the Breadth index (S) between 0-14 and 0-28 schedules are interesting; however, this conclusion on potential cross-reactive immunity must be made carefully. Given that the S compares the Ab conc for a VOC with that of the original strain, the authors should conduct a paired sample analysis instead.

As requested by the Reviewer, we have updated the analysis of Figure 4—figure supplement 1 (previously S9) to perform a paired sample analysis to further support our conclusions.

We would like to kindly thank the Reviewers and Editors for their time and effort in handling this manuscript and expect that the current revised version is suitable for publication in *eLife*.